# Best Evidence to Best Practice: Implementing an Innovative Model of Nutrition Care for Patients with Head and Neck Cancer Improves Outcomes

**DOI:** 10.3390/nu12051465

**Published:** 2020-05-19

**Authors:** Merran Findlay, Nicole M. Rankin, Tim Shaw, Kathryn White, Michael Boyer, Christopher Milross, Richard De Abreu Lourenço, Chris Brown, Gemma Collett, Philip Beale, Judith D. Bauer

**Affiliations:** 1Cancer Services, Royal Prince Alfred Hospital, Camperdown, NSW 2050, Australia; philip.beale@health.nsw.gov.au; 2Chris O’Brien Lifehouse, Camperdown, NSW 2050, Australia; michael.boyer@lh.org.au (M.B.); chris.milross@lh.org.au (C.M.); 3Sydney Catalyst Translational Cancer Research Centre, University of Sydney, Camperdown, NSW 2050, Australia; nicole.rankin@sydney.edu.au (N.M.R.); tim.shaw@sydney.edu.au (T.S.); kate.white@sydney.edu.au (K.W.); 4Research in Implementation Science and eHealth, Faculty of Health Sciences, University of Sydney, Camperdown, NSW 2006, Australia; 5Cancer Nursing Research Unit, Susan Wakil School of Nursing and Midwifery, Faculty of Medicine and Health, University of Sydney, Camperdown, NSW 2050, Australia; gemma.collett@sydney.edu.au; 6Centre for Health Economics Research and Evaluation, University of Technology Sydney, Haymarket, NSW 2000, Australia; Richard.Deabreulourenco@chere.uts.edu.au; 7National Health and Medical Research Council Clinical Trials Centre, The University of Sydney, Camperdown, NSW 2050, Australia; chris.brown@ctc.usyd.edu.au; 8School of Human Movement and Nutrition Sciences, University of Queensland, St Lucia, Brisbane, QLD 4072, Australia; j.bauer1@uq.edu.au

**Keywords:** head and neck neoplasms, malnutrition, implementation, evidence-based practice, research translation

## Abstract

Malnutrition is prevalent in patients with head and neck cancer (HNC), impacting outcomes. Despite publication of nutrition care evidence-based guidelines (EBGs), evidence–practice gaps exist. This study aimed to implement and evaluate the integration of a patient-centred, best-practice dietetic model of care into an HNC multidisciplinary team (MDT) to minimise the detrimental sequelae of malnutrition. A mixed-methods, pre–post study design was used to deliver key interventions underpinned by evidence-based implementation strategies to address identified barriers and facilitators to change at individual, team and system levels. A data audit of medical records established baseline adherence to EBGs and clinical parameters prior to implementation in a prospective cohort. Key interventions included a weekly Supportive Care-Led Pre-Treatment Clinic and a Nutrition Care Dashboard highlighting nutrition outcome data integrated into MDT meetings. Focus groups provided team-level evaluation of the new model of care. Economic analysis determined system-level impact. The baseline clinical audit (n = 98) revealed barriers including reactive nutrition care, lack of familiarity with EBGs or awareness of intensive nutrition care needs as well as infrastructure and dietetic resource limitations. Post-implementation data (n = 34) demonstrated improved process and clinical outcomes: pre-treatment dietitian assessment; use of a validated nutrition assessment tool before, during and after treatment. Patients receiving the new model of care were significantly more likely to complete prescribed radiotherapy and systemic therapy. Differences in mean percentage weight change were clinically relevant. At the system level, the new model of care avoided 3.92 unplanned admissions and related costs of $AUD121K per annum. Focus groups confirmed clear support at the multidisciplinary team level for continuing the new model of care. Implementing an evidence-based nutrition model of care in patients with HNC is feasible and can improve outcomes. Benefits of this model of care may be transferrable to other patient groups within cancer settings.

## 1. Introduction 

Malnutrition is prevalent in patients with head and neck cancer (HNC) with rates commonly reported between 30% and 50% [1]. Causes and consequences of malnutrition are multifactorial, impacting on a range of outcomes including clinical, cost and patient-centred measures for which relevant evidence is summarized in international oncology nutrition evidence-based guidelines (EBGs) [2,3,4,5,6]. As treatment for HNC has deleterious effects on organs essential for normal human functions such as eating, drinking, breathing and speaking, patients with HNC and their caregivers are among the highest in need in terms of their experience of disease and treatment burden, requiring access to well-coordinated multidisciplinary team (MDT) care before, during and after treatment [7,8]. In 2011, internationally-endorsed comprehensive EBGs for nutritional management of adult patients with HNC [9] were developed via systematic review and disseminated online via a web-based wiki platform [10]. However, evidence–practice gaps persist [11,12].

The HNC EBGs were developed according to recognised evidence synthesis methods [13,14] and the Nutrition Care Model [15,16], comprised of three domains: Appropriate Access to Care (Nutrition Screening, Nutrition Assessment); Quality Nutrition Care (Goals, Prescription, Implementation); and Nutrition Evaluation and Monitoring (Measure and Evaluate Outcomes). The best available evidence supports key recommendations for early and ongoing access to specialist oncology dietitians with expertise in complex cancer nutrition care needs, but adequate dietetic resources to deliver optimum care are frequently scarce [9,11,17,18,19,20].

Implementation of research evidence into routine clinical care is a well-recognised challenge and the focus of considerable health service research efforts in order to understand barriers to change and the most effective implementation strategies to overcome them [21]. Potential benefits of EBGs can only be realized upon successful implementation. Hence, systematic reviews highlight a range of influencing factors that require consideration in order for clinical care to align with EBG recommendations. These include complexity of the guidelines, lack of awareness or familiarity with the guidelines, lack of support at the organizational level and limitations in time and resources [22,23]. Implementation science literature broadly advocates for a multistrategic approach in order to improve adherence to EBGs in clinical practice [22,24].

To date, one Australian study has reported on the successful delivery of a dietitian-led behavior change intervention to increase adherence to the HNC EBGs, demonstrating improvements in nutritional status, quality of life and treatment completion [25,26,27]. Few other published studies have taken a multistrategic, patient-centred approach to implementing best-practice nutrition care for patients with HNC addressing barriers and facilitators at the individual, team and system levels which are then tested in a real-world setting. The aim of the present study was to: (i) determine the feasibility of implementing an evidence-based model of care for nutritional management of adult patients with HNC and; (ii) explore the subsequent impact of the new model of care on individual-, team- and system-level outcomes. Through integration with the MDT, the model of care aimed to take a patient-centred approach to delivery of nutritional care to minimise the detrimental sequelae of malnutrition and improve outcomes in this complex patient group.

## 2. Materials and Methods

### 2.1. Context and Study Design 

A mixed-methods, pre–post study design was employed to evaluate outcomes of interest both prior to, and following, implementation of a pilot evidence-based model of nutrition care in an Australian tertiary referral HNC unit. The 24 month project consisted of three phases covering pre-implementation (ten months), implementation (eight months) and analysis (six months). As the aim was to determine feasibility of delivering an evidence-based model of care, the eight month intervention phase was deemed a suitable timeframe. A clinical audit of medical records established adherence to the EBG recommendations and impact on quantitative outcomes. Exploration of barriers and facilitators to delivery of best-practice nutrition care at individual, team and system levels occurred via prior consultation with patients, caregivers [28] and clinicians. Post-implementation, team-level evaluation of the model of care involved focus groups with MDT members. This paper presents the results of the clinical audit and post-implementation MDT focus groups.

### 2.2. Study Population

In the pre-implementation and post-implementation phases, adult patients (≥18 years) undergoing radiotherapy +/− other treatment modality of curative intent for HNC were identified via the weekly MDT meeting lists. 

In the post-implementation phase, focus groups with members of the HNC MDT including medical (radiation/medical oncologists, surgeons) and specialist nursing and allied health professionals (dietitians, speech pathologists, psychologists and radiation therapists) were conducted.

### 2.3. Interventions

The pilot model of care incorporated the introduction of two key interventions as agreed to by the MDT as part of a priority setting workshop on presentation of the baseline qualitative and quantitative exploratory data collected via a retrospective clinical audit and semi-structured interviews with stakeholders. The following two interventions were ultimately chosen through reaching consensus on discussion with the MDT:

#### 2.3.1. The Supportive Care-Led Pre-Treatment Clinic

The Supportive Care-Led Pre-Treatment Clinic comprised a dedicated clinic resourced by advanced practice clinicians, specifically, a senior specialist dietitian and clinical nurse consultant to deliver structured, targeted pre-treatment assessment, intervention, education and counselling to patients and caregivers. Service structures and processes were re-organized, enabling specialist staff to deliver care aligned with EBG recommendations. Routine nursing care already provided pre-treatment assessment on an ad-hoc basis, and hence only additional senior dietetic staffing funded by the research grant was required for the pilot.

#### 2.3.2. Nutrition Care Dashboard

The Nutrition Care Dashboard served to highlight nutrition care processes and clinical outcomes and was integrated into the existing HNC Radiation Oncology list for discussion at the weekly MDT meetings. The Nutrition Care Dashboard was nominated by the MDT as a preferred intervention and was phased into meetings over several weeks through an iterative co-design process [29] to incorporate feedback from members. A key design element included a traffic light color-coded flagging system to highlight the change from baseline in both degree of critical weight loss (<5%, ≥5 to <10% and ≥10%) and nutritional status as defined by the Scored Patient-Generated Subjective Global Assessment (Scored PG-SGA) [30,31] denoting those patients who were well nourished (A), moderately malnourished (B) or severely malnourished (C). In addition, any escalation in nutrition-impact symptoms represented by the scored component of this nutrition assessment tool, validated for use in the oncology population, was also flagged.

### 2.4. Implementation Strategies

A detailed implementation plan underpinned by a recognised implementation framework [32] supported the intervention phase of this study, which included the following: clinical practice change strategies, MDT engagement, integrated care, information technology strategies, audit and feedback, staff education and support, and opinion leaders.

#### 2.4.1. Clinical Practice Change Strategies

A range of clinical practice change strategies was employed to support the systematic uptake of the EBG recommendations. These included the development of an evidence-based nutrition care pathway with clearly defined screening and assessment protocols, timing and frequency of dietetic contact before, during and after treatment and routine use of validated malnutrition screening and nutrition assessment tools. Dietitians were provided with education and support regarding use of the PG-SGA tool to ensure confidence in utilising this assessment regularly and to encourage weekly reporting of nutrition outcomes via the Nutrition Care Dashboard for discussion at the weekly MDT.

#### 2.4.2. Multidisciplinary Team Engagement

During all phases of this study, MDT members were consulted through a series of meetings with resulting feedback incorporated into the pilot model of care. Input was actively sought to co-design evidence-based nutrition care pathways and electronic clinical documentation templates, participation in relevant priority setting and education workshops, interviews and focus groups and ultimately to deliver and evaluate the model of care.

#### 2.4.3. Integrated Care

The new model of care specifically targeted key opportunities within existing care pathways and processes to facilitate uptake of the guidelines.

#### 2.4.4. Information Technology—Dietitian-Specific Clinical Documents in the Electronic Medical Record 

A suite of dietitian-specific clinical documentation templates with extractable data fields were created in the electronic medical record (eMR) to replace the free text-only documents historically available and support future sustainability through digital automation of the Nutrition Care Dashboard. As handover of clinical care occurs frequently between dietitians and care settings throughout the trajectory of patient care, the availability of standardized nutrition care information serves to enhance communication and continuity of care between clinicians and care settings.

#### 2.4.5. Audit and Feedback

The Nutrition Care Dashboard fulfilled the dual role of communicating timely clinical nutrition outcome information broadly to the MDT but also provided the function of ongoing weekly audit and feedback through highlighting whether adherence to agreed benchmarks were being met.

#### 2.4.6. Staff Education and Support

Dietitians responsible for the nutrition care of patients with HNC received practical training in administering the Scored PG-SGA and were asked to evaluate their degree of confidence in utilising the tool before and after the training session. Staff were trained in documenting in the newly created dietitian-specific clinical documents in the eMR.

#### 2.4.7. Opinion Leaders

Senior leaders within the participating cancer centre organization were invited to join the team of investigators. Their dual roles as clinical leads within the HNC MDT and also executive opinion leaders served to ensure organizational support and endorsement of the implementation program.

### 2.5. Clinical Audit Outcomes

The primary outcome in this study was the adherence to EBG nutrition care processes presented in Table 1. Adherence criteria were defined as the proportion of patients who received nutrition care in accordance with evidence-based recommendations throughout the trajectory of care. Evaluation of practice change regarding uptake of recommendations to apply validated nutrition assessment tools required a dietetic consult. The clinical audit conducted by the investigator team facilitated assessment of care delivery compared to the best available evidence. Secondary clinical outcomes included weight change (%) (calculated as difference in recorded weight (kg) between initial dietitian assessment and final week of radiotherapy), Body Mass Index (BMI, kg/m^2^) and whether radiotherapy and systemic therapy protocols were delivered as planned. System-level impact was examined via unplanned hospital admissions and dietetic resources required to deliver nutrition care. An economic analysis examined the financial implications of delivering the new model of care. Nutrition risks were captured for each patient on clinic attendance and these were categorized according to a recognised national health care quality and patient safety risk matrix [33]. Nutrition risk categorization and severity of potential incidents avoided was further verified with the organizational patient safety and quality manager.

### 2.6. Team-Level Outcomes—Multidisciplinary Team Focus Groups

Process evaluation of the model of care was undertaken according to Proctor et al.’s implementation outcomes framework [34]. MDT members were invited and consented to participate in focus groups held for each of the following disciplines: allied health and nursing professionals, oncologists, and surgeons. To ensure a group perspective was obtained [35], three semi-structured focus groups were facilitated by a study investigator (N.R.) not involved in the delivery of clinical care or the implementation strategies. The face-to-face focus groups were conducted based on a semi-structured schedule covering the same topics for each group (Appendix A). Focus groups were recorded, transcribed verbatim and thematically analysed. Two members of the research team (M.F. and N.R.) each reviewed three transcripts independently prior to meeting. Using an inductive, thematic approach, a coding framework was derived from descriptive phrases by research team members (M.F. and N.R.) who then completed coding of the transcripts. Themes were established using an iterative approach, with supporting quotations identified and extracted. 

### 2.7. System-Level Outcomes

Economic consequences of the new model of care were evaluated from an organizational perspective. For pre-implementation unplanned admissions, cost data were obtained from the hospital performance unit and are reported as the mean cost per episode in Australian dollars ($AUD). The economic analysis accounted for both direct clinical care and required associated administrative time and assumed no displacement for infrastructure or staffing resources. The latter was calculated according to the current state health professional’s award [36], inclusive of 15% on-costs.

### 2.8. Data Collection

Data were collected from hospital electronic and paper-based medical records and managed using Research Electronic Data Capture (REDCap) tools hosted at Sydney Local Health District [37].

### 2.9. Statistical Analysis

Participant baseline characteristics are reported using mean and standard deviation (SD) for continuous variables and frequency and percentage for categorical variables. Distributions of continuous variables were checked for normality and variance using the Shapiro–Wilk test and Levene’s test respectively. Normally distributed variables underwent parametric analyses while non-normally distributed data were analysed using non-parametric methods or underwent transformation to normalize data distribution. Differences in continuous variables collected pre- and post-implementation were determined via independent group t-tests or the Mann–Whitney U test, while categorical variables were analysed with Chi-squared tests. Analysis was completed using SPSS version 25 (SPSS, Chicago, IL, USA).

### 2.10. Ethics Approval and Reporting

Ethics approval was obtained from the Human Research Ethics Committee at Royal Prince Alfred Hospital, Sydney, Australia (HREC/14/RPAH/524) with site-specific approval for this study to be conducted at Royal Prince Alfred Hospital (SSA/15/RPAH/148) and Chris O’Brien Lifehouse (LH15.017). This study is reported according to the Standards for Reporting Implementation Studies (StaRI) checklist [38,39].

## 3. Results

### 3.1. Patient Characteristics

For the pre-implementation phase, a randomly selected sample of patients who commenced treatment within a two-year period (January 2013–December 2014) yielded a retrospective cohort of 100 patients who had received current standard nutrition care. One patient did not meet the inclusion criteria and one patient died prior to commencing treatment and were withdrawn from this study, yielding a final pre-implementation cohort of (n = 98). The implementation phase (October 2016 to May 2017) delivered the new model of care to a pilot cohort (n = 34). Patient demographics, diagnosis, treatment characteristics and baseline nutrition measures are summarized in Table 2. There were no differences between baseline characteristics with the exception of nutritional status, with more patients assessed as malnourished (SGA Category B/C) in the post-implementation group (n = 16, 48%) compared to those that received pre-implementation standard care (n = 12, 16%), (*p* < 0.001).

### 3.2. Adherence to Evidence-Based Guideline Recommendations

The proportion of patients that received nutrition care according to the EBG recommendations is reported for the pre-implementation and post-implementation phases in Table 3. For the domain of Appropriate Access to Care (Screening and Assessment), use of validated tools was improved for malnutrition screening 14% (n = 14 of 98) to 88% (n = 30 of 34; *p* < 0.001) and nutritional assessment (pre-treatment, *p* = 0.018 and all other time points, *p* < 0.001). Adherence to the evidence-based Dietitian Appointment Schedule was most improved at the pre-treatment time point, coinciding with the targeted Supportive Care-Led Pre-Treatment Clinic intervention, where early access to dietetic assessment and intervention increased from 20% (n = 20 of 98) to 97% (n = 33 of 34; *p* < 0.001). One patient was not offered a pre-treatment appointment as they had been unintentionally omitted from the weekly MDT list, the primary source of identifying new patients, and hence dietetic referral was not triggered. There was no improvement in adherence to the recommendation for weekly dietitian review during treatment (52% versus 59%, *p* = 0.275) or fortnightly dietetic review post-treatment (12% both groups).

### 3.3. Nutrition Outcomes

There were no statistically significant differences in BMI or mean weight change (%) between the two groups over the course of care. However, mean weight change remained below the clinically important poor prognostic threshold of ≥5% at all but one time point (−5.1% at week 6 RT) for patients who received the new model of care compared to the pre-implementation group in which progressive post-treatment weight loss was observed at post-treatment time points 1 (−6.3%), 2 (−7.5%) and 3 (−7.5%). Nutrition-related risks were identified in 74% of patients (n = 25 of 34) at presentation to the Supportive Care-Led Pre-Treatment Clinic with a median (range) of 1 (0–3) nutrition-related risk per patient (total 43 risks). These risks included malnutrition (47%), social isolation or financial hardship, suggesting either limited support and potential food and nutrition insecurity (29%), declining a clinically indicated feeding tube (9%), following an alternative, unproven diet or clinically contraindicated nutritional supplement regimen (9%), aspiration (3%) and high risk of refeeding syndrome (3%). All nutrition-related risks were categorized as potential ‘medium’ risk according to the organizational risk management framework.

### 3.4. Treatment Completion

The proportion of patients for whom radiotherapy dose was delivered as planned in the pre-implementation cohort was 89% (n = 89 of 98) versus 100% (n = 34 of 34) in the post-implementation cohort, (*p* = 0.041). Patients offered the new model of care were more likely to receive systemic therapy dose as prescribed, 67% (n = 34 of 51) versus 100% (n = 18) (*p* = 0.005). Reasons for change in treatment protocols in the pre-implementation group observed for radiotherapy were: treatment toxicity (nutrition and hydration) (2/11, 18%); treatment toxicity (other, e.g., skin reaction) (3/11, 27%); patient deterioration (1/11, 9%); other (2/11, 18%); not documented (3/11, 27%). Change in treatment protocol for chemotherapy was attributable to treatment toxicity (nutrition and hydration) (1/17, 6%) and treatment toxicity (other, e.g., myelosuppression, ototoxicity) (16/17, 94%).

### 3.5. Unplanned Admission, Economic Analysis and Dietetic Resources

There was a 15% relative (7% absolute) reduction (*p* = 0.499) in unplanned admission rates between the pre- and post-implementation groups. Reasons for unplanned admissions were not significantly different between groups, but a trend toward reduction in nutrition and hydration-related unplanned admissions was observed (20/44, 45% versus 3/13, 23%, *p* = 0.067) (Table 3). The mean cost of unplanned admissions in the pre-implementation cohort was $AUD30,897/episode. At the system level, the new model of care avoided a projected annualized 3.92 unplanned admissions and related expenses of $AUD121,100/annum. In terms of dietetic occasions of service (OOS), the new model of care did not require more dietetic resources to deliver (*p* = 0.613). Patients who required unplanned hospital admission utilised more dietetic resources compared to those who were not admitted over the course of care (median (range) OOS 15 (3–66) versus 8 (2–29), *p* < 0.001). Taking into account the investment required to resource a senior dietitian to deliver the new model of care, there remained a cost reduction of $AUD14.65 for every dollar spent on delivering the new model of care.

### 3.6. Fidelity

Fidelity was monitored to ensure the intervention was delivered as planned during the study period between October 2016 and May 2017. For the Supportive Care-Led Pre-Treatment Clinic, 97% (n = 33 of 34) of eligible patients were offered a clinic appointment and 100% of these attended. For the Nutrition Care Dashboard integrated into the weekly radiotherapy list, discussion occurred as intended at 100% of MDT meetings during the intervention period whilst the MDT clinical year was in session.

### 3.7. Contextual Changes

Substantial organizational change occurred between the pre-implementation and post-implementation study phases. During the study period, cancer services within which the head and neck oncology unit sits transitioned from a large public hospital to a public–private partnership model of service delivery. While treatment services remained largely unchanged, challenges arose with the introduction of separate allied health staff designated for public and privately insured patients. Every effort was made to take an inclusive approach to the range of participating clinicians providing care to patients with HNC regardless of employing organization. While it was not possible to control for the scale of organizational change, the authors acknowledge that this occurred during the study period and believe this to be reflective of conducting implementation research in a real-world setting and that the flexibility and adaptability of the research team ensured any issues that arose were largely able to be addressed.

### 3.8. Harms

No harms were identified as a result of the implementation. Conversely, as presented elsewhere in this paper, the new model of care facilitated early identification of a range of nutrition risks that were addressed sooner, potentially minimising the risk of any nutrition-related harm to patients during their cancer care.

### 3.9. Implementation of Team-Level Evaluation—Multidisciplinary Focus Groups

MDT members (n = 12) participated in a series of three focus groups with allied health (n = 5), medical (n = 5) and nursing (n = 2) disciplines represented, confirming clear support for continuing the new model of care. The key themes identified and supporting quotations are presented in Table 4. To preserve participant anonymity, allied health professional and nursing discipline responses are reported as “Supportive Care Clinician” with oncologists and surgeons reported as “Medical Clinician.” 

## 4. Discussion

This is the first study to: (i) investigate the feasibility of delivering an evidence-based model of nutrition care for patients with HNC utilising supportive care and e-health technology interventions and; (ii) explore the subsequent impact of the new model of care at individual, team and system levels. The key interventions supported by multicomponent implementation strategies resulted in improvements in adherence to EBG recommendations for nutrition care. This directly contributed to clinically relevant reductions in critical weight loss, greater treatment completion rates and economically pertinent decreases in unplanned admissions and lower costs of care. Although this study was not powered to detect a statistically significant different in weight as a secondary outcome, these remain promising findings, particularly given the higher prevalence of malnutrition at baseline in the post-implementation cohort.

The findings highlight that, with a coordinated approach and adequate support, practice change in alignment with the best available evidence is possible. This holds important implications for health services aiming to deliver high-value care to complex patients with HNC. Although it has been recognised that the evidence base to support decision making regarding optimal guideline implementation strategies is imperfect [40], tailored interventions and adequate support are acknowledged to be key to successful guideline implementation [41]. Our study corroborates findings of other investigators who also found that dietitian-led behavior change intervention improved uptake of nutrition care guidelines in patients with HNC [26].

A review of guideline implementation strategies found a 10% absolute improvement in the desired direction of practitioner behavior change [42], whereas our study demonstrated an absolute increase in adherence to nutrition care process recommendations for: malnutrition screening with a validated tool (74%) and; nutrition assessment with a validated tool (15% at initial assessment, 76% during radiotherapy, 48% post-radiotherapy). In terms of adherence to the Dietitian Appointment Schedule, the greatest improvement was seen in early access to care with the introduction of the targeted intervention in the form of the Supportive Care-Led Pre-Treatment clinic (20% versus 97%). This may represent an increase in the on-treatment clinic capacity with resource-intensive initial consultations, particularly for patients requiring education and counselling regarding feeding tubes, allocated to a dedicated pre-treatment assessment. The absence of improvement in access to dietetic care post-treatment is likely attributable to clinic demand exceeding capacity and thus a system and resourcing issue, findings that are corroborated by McCarter et al. (2018). Based on the pilot results, we hypothesize that were similar targeted interventions to be applied elsewhere in the patient care pathway, greater guideline uptake and further improvements in outcomes may be observed.

To our knowledge, this is the first study to test the implementation of a Nutrition Care Dashboard as a decision support tool for MDTs in patients with head and neck cancer. Health care dashboards are gaining recognition in their ability provide streamlined data collection and user-friendly visualization to quickly and efficiently guide clinical decision making [43]. Key elements of quality decision-making tools include that they provide ready access to timely, relevant and reliable data that incorporate trends or benchmarks [44]. Strategies for automation of the dashboard through creation of extractable data fields in the eMR have laid the foundation for future sustainability of this tool for use within the MDT. Our study demonstrated strong support for continuation of the dashboard at the team level due to the enhanced communication and ability to flag the need for a coordinated MDT response to the deteriorating patient.

The United Kingdom’s National Institute for Clinical Excellence [45] published guidance regarding structure of services required to deliver care to complex patients with HNC highlights the importance of expert dietitians as core members of the MDT [7,8]. Despite this, adequate dietetic resourcing to deliver best practice nutrition care in cancer services is frequently met with financial constraints. In our study, patients who received the new model of care experienced clinically relevant fewer hospital admissions and those managed in the ambulatory setting used fewer dietetic resources overall, providing support for a proactive approach to nutrition care. The new model of care resulted in reduced costs of care equating to employing one full time equivalent (FTE) senior allied health clinician per annum. In terms of sustainability, this justifies the investment required for continuing the new model of care from an economic viewpoint. Additionally, the benefits to clinical outcomes and improved access to care, care coordination and communication suggest an evidence-based model of nutrition care can contribute to high-quality, high-value service delivery for these complex patients. Another important element of sustainability for the new model of care identified in our study includes leadership, both within dietetics and within the MDT more broadly. Greater awareness of the value of cancer nutrition care amongst health service administrators and policy makers is also required in order to optimized outcomes through research translation.

Staff and system domains should be considered as active components of the change process in order to develop feasible and acceptable interventions in health care settings [46]. Clear support for continuing the new model of care was evident at the team level from MDT member focus group data. Improved care processes, workflow and time management, clinical leadership within the MDT, value of nutrition care, integrated care coordination and communication and patients being prepared for care were all identified as key elements or benefits of the new model of care. These data suggest that the new model of care was considered by MDT members to be feasible, acceptable and appropriate.

This study has a number of strengths including the patient, caregiver and clinical team-centred design of the new model of care, high fidelity to the intervention strategies and sound consultation with and participation rates by the target groups. As this was a pilot study testing the feasibility of delivering an evidence-based model of nutrition care, the authors acknowledge that participant numbers are small. As a result, confirmation of these preliminary findings in a larger patient cohort is warranted. The model of care could further be enhanced through exploration of service innovations that reduce appointment burden and improve access to follow-up care in line with EBG recommendations, particularly for long-distance patients [28]. Novel allied health-led telemedicine models of care have demonstrated improved service efficiency and treatment satisfaction for patients with HNC [47,48] and hold particular relevance in Australia and other countries with geographically vast services. Other study limitations include the inability to blind the MDT to the new model of care, which may have influenced clinician motivation to adhere to the EBG recommendations. There were significant organizational changes during the study period, although the research and clinical teams endeavored to mitigate these challenges wherever possible.

## 5. Conclusions

This pilot study demonstrates that delivering a patient-centred, evidence-based model of nutrition care in a major HNC centre is feasible and contributes to improved nutrition care processes and clinical and cost outcomes. From an economic analysis point of view, the new model of care resulted in lower costs of care. Benefits were also experienced by the clinical team through enhanced coordination, communication and integration of care. These findings hold important implications for the provision of evidence-based nutrition care to patients with HNC, suggesting that dietetic-led interventions can deliver high-value care.

## Figures and Tables

**Table 1 nutrients-12-01465-t001:** Evidence-based guideline recommendations and adherence criteria.

Nutrition Care Framework	Recommendation	NHMRC ^a^ Grade	Adherence Criteria
Access to Care Screening and Assessment	• Malnutrition screening should be undertaken on all patients at diagnosis to identify nutritional risk and then repeated at intervals through each stage of treatment (e.g., surgery, radio/chemotherapy and post-treatment).	B	• Screening using the MST ^b^ occurred before Week 1 of radiotherapy.
	• All patients receiving radiotherapy to the head and neck should be referred to the dietitian for nutrition support.	B	• Dietetic consult occurred before Week 1 of radiotherapy.
	• Use a validated nutrition screening tool (e.g., MST) for identifying malnutrition risk.	B	• Use of the MST occurred before Week 1 of radiotherapy.
	• Use a validated nutrition assessment tool (e.g., PG-SGA ^c^).	B	• Use of PG-SGA occurred when assessing nutritional status.
Quality Nutrition Care	• Weekly dietitian contact improves outcomes in patients receiving radiotherapy.	A	• Dietetic consult occurred for every five fractions of radiotherapy given in a single working week period.
Nutrition Monitoring and Evaluation	• Patients should be seen weekly by a dietitian during radiotherapy.	A	• As above.
	• Patients should receive minimum fortnightly follow up by a dietitian for at least 6 weeks post-treatment.	A	• Dietetic consult occurred at least once in a 14 day period following end of radiotherapy for three consecutive fortnights.
	• Monitor weight, intake and nutritional status during and post-(chemo)radiotherapy.	A	• Use of Scored PG-SGA occurred at baseline, mid-RT ^d^ (Week 3–4), end-RT (Week 6–7) and at post-RT dietitian consults.

^a^ NHMRC = National Health and Medical Research Council; ^b^ MST = Malnutrition Screening Tool; ^c^ PG-SGA = Patient-Generated Subjective Global Assessment; ^d^ RT = radiotherapy.

**Table 2 nutrients-12-01465-t002:** Comparison of clinical characteristics between pre- and post-implementation cohorts.

Characteristic		Pre-Implementation (N = 98)			Post-Implementation (N = 34)			*p* Value *
		N		(%)	N		(%)	
Age, Years								0.394 **
	Mean (SD)		61.8 (12.3)			63.8 (10.4)		
Gender								0.281
	Male	75		(77)	29		(85)	
	Female	23		(23)	5		(15)	
Disease Stage								0.935
	I	2		(2)	1		(3)	
	II	12		(14)	3		(10)	
	III	17		(20)	7		(24)	
	IV	55		(64)	18		(62)	
Tumour Site								0.719
	Oral cavity/lip	18		(18)	6		(18)	
	Oropharynx	36		(37)	18		(53)	
	Hypopharynx	3		(3)	1		(3)	
	Larynx	13		(13)	4		(12)	
	Nasopharynx	15		(15)	1		(3)	
	Nasal/paranasal sinus	3		(3)	1		(3)	
	Salivary gland	7		(7)	2		(6)	
	Other/unknown primary	3		(3)	1		(3)	
Tumour Type								0.117
	Squamous cell carcinoma	86		(88)	33		(97)	
	Other	12		(12)	1		(3)	
Treatment Modality								0.361
	RT ^a^—definitive	15		(15)	9		(26)	
	CRT ^b^—definitive	43		(44)	16		(47)	
	Surgery + CRT—adjuvant	8		(8)	2		(6)	
	Surgery + RT—adjuvant	32		(33)	7		(21)	
Performance Status								0.310
	ECOG ^c^ 0	41		(42)	17		(50)	
	ECOG 1	32		(33)	14		(41)	
	ECOG 2	6		(6)	0		(0)	
	ECOG 3	1		(1)	0		(0)	
	ECOG 4	0		(0)	0		(0)	
	Not documented	18		(18)	3		(9)	
Tobacco Use								0.143
	No	33		(34)	15		(44)	
	Yes	56		(57)	19		(56)	
	Not documented	9		(9)	0		(0)	
Smoking Status								0.133
	Never smoked	0		(0)	0		(0)	
	Current smoker	19		(34)	3		(16)	
	Previous smoker	37		(66)	16		(84)	
Alcohol Use								0.096
	None or social only	49		(50)	21		(62)	
	1–2 standard drinks/d	8		(8)	2		(6)	
	>2 standard drinks/d	26		(27)	11		(32)	
	Not documented	15		(15)	0		(0)	
HPV ^d^ Status								0.059
	Negative	4		(4)	5		(15)	
	Positive	17		(17)	8		(24)	
	Not documented	77		(79)	21		(62)	
Nutrition Support Delivery Mode								0.852
	Gastrostomy—PEG ^e^	37		(54)	10		(50)	
	Gastrostomy—RIG ^f^	13		(19)	4		(20)	
	Gastrostomy—surgical	2		(3)	0		(0)	
	NGT ^g^	15		(22)	6		(30)	
	TPN ^h^	1		(1)	0		(0)	
Height, cm								0.768 **
	Mean (SD)		171.5 (8.5)			172.0 (9.1)		
Weight, kg								0.954 **
	Mean (SD)		75.6 (23.0)			75.3 (19.4)		
BMI ^i^, kg/m^2^								0.824 **
	Mean (SD)		25.5 (7.1)			25.2 (5.4)		
Nutritional Status, PG-SGA ^j^ Score								<0.001 **
	Mean (SD)		4.4 (5.6)			8.7 (4.5)		
Nutritional Status, PG-SGA Category								<0.001
	A (well nourished)	63		(84)	17		(52)	
	B (moderately malnourished)	8		(11)	13		(39)	
	C (severely malnourished)	4		(5)	3		(9)	

^a^ RT = radiotherapy; ^b^ CRT = chemoradiotherapy; ^c^ ECOG = European Co-Operative Group; ^d^ HPV = Human Papilloma Virus; ^e^ PEG = Percutaneous Endoscopic Gastrostomy; ^f^ RIG = Radiologically Inserted Gastrostomy; ^g^ NGT = Nasogastric Tube; ^h^ TPN = Total Parenteral Nutrition; ^i^ BMI = Body Mass Index; ^j^ PG-SGA = Patient-Generated Subjective Global Assessment; * χ^2^ of independence; ** *t*-test.

**Table 3 nutrients-12-01465-t003:** Summary of key results—comparison of outcome measures between pre- and post-implementation cohorts.

Outcome	Measure/NHMRC ^a^ Grade of Recommendation		Pre-Implementation (N = 98)			Post-Implementation (N = 34)			*p* Value *
			N		(%)	N		(%)	
Process	Nutrition screening (Grade B)								<0.001 **
		Screened with validated tool	14		(14)	30		(88)	
	Nutritional assessment (Grade B)								
		Nutritional assessment with validated tool on dietitian review							
		- pre-treatment	73		(85)	33 of 33		(100)	0.018 **
		- during treatment	3		(3)	26 of 34		(79)	<0.001 **
		- end treatment	5		(6)	15 of 34		(54)	<0.001 **
		- post-treatment (T1)	2		(3)	22 of 29		(73)	<0.001 **
		- post-treatment (T2)	3		(6)	14 of 20		(67)	<0.001 **
		- post-treatment (T3)	3		(9)	10 of 17		(59)	<0.001 **
	Dietitian Appointment Schedule (Grade A)								
		Received recommended dietitian assessment							
		- pre-treatment	20		(20)	33		(97)	<0.001 **
		- weekly during treatment	47		(48)	20		(59)	0.275 **
		- fortnightly for 6 weeks post-treatment	12		(12)	4		(12)	0.864 **
Clinical	Radiotherapy delivered as planned								0.041
		No	11		(11)	0		(0)	
		Yes	87		(89)	34		(100)	
	Systemic therapy delivered as planned								0.005
		No	17		(33)	0		(0)	
		Yes	34		(67)	18		(100)	
	Weight change during treatment, %								0.432 **
		Mean (SD)		−5.9 (4.2)			−4.6 (5.3)		
	BMI ^b^ post-treatment, kg/m^2^								0.989 **
		- Mean (SD)		24.3 (5.6)			24.3 (4.1)		
System	Dietitian resources—occasions of service								0.613 **
		Mean (SD)		13.1 (9.8)			14.1 (7.1)		
	Unplanned admission								0.499
		No	54		(55)	21		(62)	
		Yes	44		(45)	13		(38)	
	Unplanned admission—reason								0.067
		Treatment toxicity—nutrition/hydration	20		(45)	3		(23)	
		Treatment toxicity—other	14		(32)	7		(54)	
		Social circumstances	3		(7)	3		(23)	
		Other	7		(16)	0		(0)	

^a^ NHMRC = National Health and Medical Research Council; ^b^ BMI—Body Mass Index; * χ^2^ of independence; ** *t*-test.

**Table 4 nutrients-12-01465-t004:** Qualitative analysis—key themes identified through focus groups with head and neck oncology multidisciplinary team members (N = 12).

Theme	Supporting Qualitative Data
**Improved Process, Workflow and Time Management**	“…that list (Nutrition Care Dashboard) works like our bible in terms of who’s on treatment, where they’re at and what’s going on.”
	*Supportive Care Clinician 6*
	“My clinic on the Monday morning now runs on time, because I don’t spend an hour and a quarter with them…. I can spend (my time) … talking about the treatment, the radiotherapy...”
	*Medical Clinician 3*
**Clinical Leadership Within the MDT**	“Well we definitely have a structure to the MDT. I do not think the structure extends well beyond surgery and radiation therapy. I think that leadership outside of that should be allocated, because at the moment it is really just assumed…I think that it would be very beneficial for us to have a well-established structure, as to how the service is run, who answers to whom, and who is control of what. I think a lot of it is assumed and really should be actually spelt out.”
	*Medical Clinician 5*
	“There needs to be a driver. A champion…and someone to be present at the MDT because the list comes up and the doctors look around, and it’s like someone needs to start talking.”
	*Supportive Care Clinician 1*
**Value of Nutrition Care**	“We are less likely to lose people. But what the intangible is that the intake - what I call intake - the pre-therapy assessment forces the multidisciplinary team, particularly the surgeons, to stop and think about the radiotherapy.”
	*Medical Clinician 3*
	“And it’s (nutrition care) not really my expertise, so as a dietitian, I would do a worse job and cost more to do it.”
	*Medical Clinician 3*
	“I think there are two aspects that struck me, which were it’s useful to know what the effect of the surgery has been, going into radiation therapy...So to see the percentage body weight loss was educational. It is also useful to have a comparison…because as surgeons we do not really know what the typical nutritional effects of radiation therapy are. We have our assumptions and biases but we do not really have any objective evidence.”
	*Medical Clinician 4*
**Integrated Care Coordination and Communication**	“Well, so it’s a set formal structured clinic where patients are seen pre-radiotherapy and it’s at a time set aside with the nurse and the dietitian and the patient attends that sole appointment…Whereas prior to that, I would have to try and catch them which was very haphazard and I don’t believe that the patients were concentrating on our consult or our education. This (Nutrition Care Dashboard) is like a checklist now, it’s an assurance that patients are educated particularly those having a gastrostomy tube at a point in time prior to treatment where they can absorb the information. If they don’t understand the information they can contact us...So, it’s a thorough process– I’ve set aside time now in my weekly routine that I attend this clinic on a Wednesday morning. Rather than I’ll go Monday, I’ll go Tuesday I’ll go and try and find them here. It is much easier - structured I guess, for patients and for me.”
	*Supportive Care Clinician 7*
	“And it’s (the Nutrition Care Dashboard) a visual support tool…as a team to say, “This is why we need such a strong Allied Health team because look at all the patients that you’re looking after. It’s not just you’re on treatment, see you later. It’s an ongoing care.”
	*Supportive Care Clinician 4*
**Prepared for Care**	“The pre-treatment clinic provides them (patients and caregivers) the information and the dedicated environment which is not the same as when they’re getting told about their radiation and their diagnosis….”
	*Medical Clinician 1*
	“…something I am no longer surprised when I’m asked - when a patient needs admission during radiotherapy because now we usually we see it coming.”
	*Medical Clinician 3*
	“I would say, at the moment, it feels really good that patients know what they’re doing, where they’re going, not all new and scary information when I’m first seeing them which is amazing. It’s great. You don’t have to go through everything because they’ve found that information out previously. They’ve absorbed it, they’re ready for it, so that means they’re ready for the next lot of information that they need through their radiotherapy.”
	*Supportive Care Clinician 5*

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
