# Peer review of "Best Evidence to Best Practice: Implementing an Innovative Model of Nutrition Care for Patients with Head and Neck Cancer Improves Outcomes"

_nutrients, 2020, doi:10.3390/nu12051465_

Round 1
Reviewer 1 Report
This is an interesting study demonstrating the impact of a dashboard to support practice change in nutrition in head and neck cancer. Can you comment on why the post implementation "n" is smaller than the pre-implementation "n" and how you justified this. Also, can you comment on why there was not change in weight given enhanced compliance to nutrition care.
Reviewer 2 Report
Patients with head and neck cancer usually require unplanned admission resulting from treatment-related toxicities and interrupt the radiotherapy and systemic therapy. In Table 3, data shows a 100% of the clinical outcomes of radiotherapy and systemic therapy delivered rate as planned and a 38% (13/34) of unplanned admission rate in post-implementation cohort. Despite there was a 15% relative (7% absolute) reduction (p=0.499) in unplanned admission rates between the pre- and post-implementation groups, the reasons of unplanned admission and the mechanism of whether radiotherapy and systemic therapy protocols could be delivered as planned should be addressed in Results.
Reviewer 3 Report
The study aim is to implement an evidence-based strategy for nutritional management in patients with head and neck cancer and evaluate its effect on clinical and system level outcomes. Evidence-based guidelines on the nutritional management in head and neck cancer are available internationally, however, few studies have addressed how these guidelines can be properly implemented in a clinical setting. Therefore, the present study contributes with important knowledge on successful implementation of nutritional guidelines within the multidisciplinary team and the effect on clinical outcome.
Broad comments
Abstract
In the abstract, the authors describe a baseline clinical audit for n = 98. However, the results presented in the abstract from this audit does not match the results in the manuscript.
Introduction
The introduction is well established in previous research, addressing nutritional guidelines from the leading international nutritional organizations.
Aim
The aim of the present study was to: ii) explore the subsequent impact of the new model of care on clinical and cost outcomes which corresponds well with the clinical Audit Outcomes described in section 2.5. In the Discussion, the study is outlined to explore the ii) subsequent impact at individual, team and system levels.
- It is not clear for the reader what the authors mean by the ‘team-level’ and, therefore, it is unclear if the study particularly addresses the ‘team-level’ aspects in the results.
- However, the study do present System Level Outcomes, as defined by the present study, which is not properly addressed in the study aim.
Materials and Methods
The study is well designed and include a number of parallel processes. It is evident that the authors have made their uttermost to guide the reader through the different steps of the methods used and data collection. However, some issues still needs to be further addressed.
The study included two key interventions, namely, a Supportive Care-Led Pre-Treatment Clinic and the Nutrition Care Dashboard. However, it is not clear to the reader how the ‘clinical practice change strategies’ or the ‘dietitian-specific clinical documents’ described in page 4 align with these two interventions.
Please provide the reader with more information on the focus groups, namely, numbers completed, persons involved in each group, topics used for group discussions etc. What was the purpose of the face-to-face interviews vs. focus groups? Were data from the face-to-face interviews and focus groups analysed separately?
Statistical analyses:
- Considering the small amount of patients in some of the groups, was the Chi-squared test appropriate to use for all analyses?
- It is not clear to the reader what multivariate analyses that were used and when.
Results
The authors declare that there were no statistically significant differences in mean weight change (%) between the two groups over the course of care. To be able to put these results into a clinical perspective, the authors needs to give the reader information on how the weight loss (%) was calculated, and when, and reference weight used – preferable in the methods-section. Was weight measured in a clinical setting or self-assessed by the patient?
The authors also write that ‘mean weight change remained below the clinically important poor prognostic threshold of ≥5% all but one time point (-5.1% at week 6 RT) for patients who received the new model of care’. It would be useful for the reader to be presented with the same data for the pre-implementation group to be able to put this result into perspective. This is importance since the authors particularly highlights this result as one of the main outcomes of the study.
Qualitative analysis
- It is not clear in the results section how data from the focus groups and face-to-face interviews were used, together or separately. In the result section, the research group only talks about the focus groups.
- Allied health and nursing disciplines are not represented in the supporting quotations presented in Table 4, making the result seem a bit skewed since only the perspectives from the clinicians are presented.
- It is not clear to the reader how the themes concedes with the new model of nutritional care, it seems like the MDT members speak in general matters, not specifically of the implemented nutritional regime. This may be clearer to the reader if the process behind the focus groups are more properly presented in the methods section i.e. topics used for group discussions.
Discussion
The authors describes in line 382-383 that the new model of care was considered feasible, acceptable and appropriate by the MDT members. Has this been previously presented in the result-section?
The new process of care is well known by the MDT members and, as highlighted earlier, the study is well designed and include a number of parallel processes. This may have influenced the possibility to implementation of the model of nutritional care into clinical practice i.e. made the MDT members keener to keep-up the new structures. Therefore, an important aspect to discuss is how to reach sustainability over time for the new model of care. The authors could also discuss on changes that would improve the new model of care.
Conclusion
Uncertainties exists on whether the MDT members talks specifically of the new model of nutritional care during the focus-groups and, therefore, it is unclear if this part of the conclusion is justified and supported by the results.
Specific comments
Table 2 & 3: Please specify the statistical method(s) used in a footnote below the table.
Table 2: Tumour type – Other (please specify), it seems like something has been missed in this section.
Table 3: Please look at the layout of the table legend.
Table 3: Please specify what is meant by ‘BMI change during treatment’.
Table 4: first quote, if the supportive care clinician 6 mean the Nutrition Care Dashboard when talking about the ‘list’, this can be clearer to the reader using a bracket where this could be specified.
Reviewer 4 Report
Very interesting study and well-written paper. I have included a few comments below:
Methods
Page 3, Line 99: By consumers do you mean patients?
Page 3, Line 113: Could you expand a bit on what the baseline exploratory data consisted of and found and how these two interventions were ultimately chosen
Page 4, Line 147: Typo: “Input was actively sort to co-design”
Page 5, Section 2.7: Once coding book was developed who then coded the manuscripts
Focus groups: can you be more clear on how many focus groups and whether these were structured, semi-structured or unstructured
Results
Page 7, Line 237: “Patient, diagnosis treatment characteristics”, I think this should read “Patient diagnosis, treatment characteristics”
Page 11, Line 285: Can you give more information, probably in the methodology as to how exactly these costs were calculated
Discussion
Line 337: You say this is the first study but in your introduction you mention one other study integrating these in Australia?
Reviewer 5 Report
Congratulations. Good and needed paper.
